# Data Mining Identifies *CCN2* and *THBS1* as Biomarker Candidates for Cardiac Hypertrophy

**DOI:** 10.3390/life12050726

**Published:** 2022-05-12

**Authors:** Markus Johansson, Benyapa Tangruksa, Sepideh Heydarkhan-Hagvall, Anders Jeppsson, Peter Sartipy, Jane Synnergren

**Affiliations:** 1Systems Biology Research Center, School of Bioscience, University of Skövde, SE-541 28 Skövde, Sweden; sepideh.heydarkhan.hagvall@his.se (S.H.-H.); peter.sartipy@his.se (P.S.); jane.synnergren@his.se (J.S.); 2Department of Molecular and Clinical Medicine, Institute of Medicine, The Sahlgrenska Academy at University of Gothenburg, SE-413 45 Gothenburg, Sweden; anders.jeppsson@vgregion.se; 3Bioscience, Research and Early Development, Cardiovascular, Renal and Metabolism (CVRM), BioPharmaceuticals R&D, AstraZeneca, SE-413 83 Gothenburg, Sweden; 4Department of Cardiothoracic Surgery, Sahlgrenska University Hospital, SE-413 45 Gothenburg, Sweden

**Keywords:** cardiac hypertrophy, stem cells, biomarker, endothelin-1, transcriptomics, disease model

## Abstract

Cardiac hypertrophy is a condition that may contribute to the development of heart failure. In this study, we compare the gene-expression patterns of our in vitro stem-cell-based cardiac hypertrophy model with the gene expression of biopsies collected from hypertrophic human hearts. Twenty-five differentially expressed genes (DEGs) from both groups were identified and the expression of selected corresponding secreted proteins were validated using ELISA and Western blot. Several biomarkers, including *CCN2*, *THBS1*, *NPPA*, and *NPPB*, were identified, which showed significant overexpressions in the hypertrophic samples in both the cardiac biopsies and in the endothelin-1-treated cells, both at gene and protein levels. The protein-interaction network analysis revealed *CCN2* as a central node among the 25 overlapping DEGs, suggesting that this gene might play an important role in the development of cardiac hypertrophy. GO-enrichment analysis of the 25 DEGs revealed many biological processes associated with cardiac function and the development of cardiac hypertrophy. In conclusion, we identified important similarities between ET-1-stimulated human-stem-cell-derived cardiomyocytes and human hypertrophic cardiac tissue. Novel putative cardiac hypertrophy biomarkers were identified and validated on the protein level, lending support for further investigations to assess their potential for future clinical applications.

## 1. Introduction

Cardiac hypertrophy is a condition in which the heart muscle thickens as an adaptive response to several stimuli. When stimulated, cardiomyocytes (CMs) increase in size and cause an enlargement of the heart [1], which can be further categorized into physiological or pathological cardiac hypertrophy. Physiological hypertrophy occurs naturally by exercise, pregnancy, or puberty [2,3,4]. This type of hypertrophy is generally harmless and reversible. On the other hand, prolonged pathological cardiac hypertrophy can lead to heart failure and severe cardiovascular disease. The development of pathological cardiac hypertrophy is associated with aortic stenosis, gene mutations, chronic hypertension, neurohormonal stimulation, and physical stretching [5,6]. The heart increases in size and is unable to pump blood effectively due to the remodeling of several pathways that affect the morphology and function of the heart. The characteristics of pathological cardiac hypertrophy include an increase in glucose consumption by the CMs, an increase in natriuretic peptide protein production, and an aberrant gene-expression profile [7,8,9].

To study cardiac hypertrophy in vitro, model systems of high quality and high human relevance are needed. We and others have reported on the development of hypertrophic disease models based on human-induced pluripotent stem-cell-derived CMs (hiPSC-CMs), which provides advantages in terms of high throughput and high translatability of the results [10,11,12].

To induce hypertrophy in hiPSC-CMs, endothelin 1 (ET-1) stimulation or stretch stimulation was applied [10,13,14,15,16]. ET-1 is a neurohormone that contributes to the thickening of heart muscles by stimulating receptors that couple to the heterotrimeric G-protein. The stimulation results in the activation of downstream signaling pathways that lead to an increase in catalytic activities and trigger hypertrophic conditions [15]. ET-1 stimulation in cell cultures increases the size of the CMs and increases the lactate levels due to higher glucose consumption. It also results in elevated levels of A- and B-type natriuretic peptides (ANP and BNP) [10,11,16,17], which are characteristics of pathological cardiac hypertrophy.

The similarities between in vitro cardiac hypertrophy models and the in vivo conditions in human hypertrophic hearts are of high importance for the translational relevance of the models. Aggarwal et al. (2014) previously reported on a comparative study of differentially expressed genes (DEGs) in human myocardial biopsies from patients with aortic stenosis-induced left ventricular hypertrophy with DEGs of an ET-1 stimulated hiPSC-CMs hypertrophic model using a principal component analysis approach. Their heart biopsies were obtained from patients with cardiac hypertrophy with different levels of ejection fractions (EFs). Interestingly, a large number of genes with similarities in expression changes in all hypertrophic samples were identified [12]. However, further analysis of the correlations between mRNA profiles of in vitro cardiac hypertrophy models and corresponding in vivo cardiac tissue is needed.

In the present study, we compare the overlap of DEGs identified in data from our in vitro hiPSC-CM hypertrophy model [14] and compare this to the transcriptional data from patient cardiac biopsies to identify biomarkers for the diagnosis and monitoring of cardiac hypertrophy. Several candidates were identified and some of these were selected for further validation of protein levels.

## 2. Materials and Methods

### 2.1. Transcriptomics Datasets from In Vitro and In Vivo Experiments

Gene-expression data from hypertrophic hiPSC-CMs [10] were used in this study. Briefly, the hypertrophic response was induced in the hiPSC-CMs by ET-1 stimulation (10 nM). Every 24 h, fresh medium containing ET-1 was added to the hiPSC-CMs. RNA-seq was carried out in triplicates on the ET-1-stimulated and -untreated cells after 8, 24, 48, 72, and 96 h, respectively. Johansson et al. reported that 93% of CMs were positive for the cardiac specific marker cardiac troponin T [10]. In the study, DEGs were identified by the quasi-likelihood F-test using the *edgeR* package. DEGs with a false discovery rate (FDR) ≤ 0.05 and an absolute fold change (FC) value of ≤2 were considered as differentially expressed. We combined the identified DEGs from all time points in our downstream analysis. The raw and processed RNA-seq datasets are accessible at ArrayExpress (https://www.ebi.ac.uk/arrayexpress/, accessed on 4 April 2022); accession number: E-MTAB-8548.

To investigate the correlations between the in vitro hypertrophy model and corresponding human hypertrophic cardiac tissue, data from myocardial left ventricular biopsy samples from a publicly available microarray dataset [12] were analyzed. This dataset contained two experimental groups and one control group. The experimental groups were left ventricular samples from male patients with aortic stenosis and noticeable left ventricular hypertrophy, in which EF was classified as normal (EF > 50%, n = 3) or low (EF < 30%, n = 3). The control biopsies were collected from male patients with coronary disease, but without a hypertrophic phenotype and with normal EF (>60%, n = 3). The RNA isolated from biopsies was analyzed using Affymetrix HG-U133A and U133B microarrays. This microarray data is accessible at ArrayExpress (https://www.ebi.ac.uk/arrayexpress/, accessed on 4 April 2022); accession number: E-MEXP-2296.

### 2.2. Differentially Expressed Genes in Biopsies from Cardiac Hypertrophic Biopsies

The gene-expression data from the cardiac biopsies [12] was analyzed to identify DEGs using the *Miodin* R package [18]. The background filtering threshold was set to six to remove low-expressed transcripts. The background filtering threshold was selected based on the average expression value of pluripotency marker genes (*LIN28A*, *NANOG*, *PF4*, and *POU5F1*), which are not expected to be expressed in differentiated cells [19,20]. The normalized data from U133A GeneChip contained 22,283 features. After the background filtering, 16,017 and 16,047 features remained for low and normal EF samples, respectively. For the U133B GeneChip, the normalized data contained 22,645 features. After the background expression filtering, 12,339 and 12,293 features remained for low and normal EF samples, respectively. The genes with an absolute log_2_FC of >1 and FDR ≤ 0.05 were considered as statistically significant. The DEGs identified from low and normal EF samples were combined for downstream analysis.

### 2.3. Functional Annotation Analysis of Differentially Expressed Genes Using IPA

The DEGs from all samples were uploaded to IPA for functional annotation. Duplicates and ambiguous transcripts that represented multiple loci or genes were removed. In total, 1496 unique DEGs from the in vitro data and 126 unique DEGs from the in vivo cardiac biopsies data were analyzed with the use of Ingenuity Pathway Analysis (IPA) software (QIAGEN Inc., Redwood City, CA, USA, https://digitalinsights.qiagen.com/IPA). In total, 25 overlapping DEGs between the heart biopsies and the cardiac stem-cell-based model were identified. The Core Analysis and Tox functions in IPA were applied for a functional assessment of the biomarkers and their subcellular locations, cardiac clinical endpoints, associated pathways, and significant cardiac hypertrophy associations (*p* ≤ 0.05) were identified. The pathways associated with more than two DEGs were reported in this analysis. Based on the subcellular locations and detectability, seven upregulated genes, for which corresponding proteins are secreted into the extracellular space and are detectable in biofluids, were selected for further protein validation. Figure 1 shows the workflow for candidate biomarker identification, functional assessment, and the protein-validation process.

### 2.4. Protein Validation of Selected Cardiac Hypertrophy Candidates

For the analysis of the secreted proteins, conditioned media were collected in triplicates and centrifuged (300× *g*, 3 min) to remove potential cell debris, and stored at −80 °C. ELISA was used to measure the concentration of ANP (cat EIAANP, ThermoFisher Scientific, Waltham, MA, USA), proBNP (cat EHPRONPPB, ThermoFisher Scientific), and *CCN2* (*CTGF*) (cat ab261851, Abcam, Cambridge, United Kingdom), according to the manufacturer’s instructions. Due to difficulties of analyzing *THBS1* and NES using ELISA, quantitative Western blot was employed for those specific proteins.

A more detailed description is included in the Appendix A. In short, the prepared samples were loaded onto a NuPAGE™ 4–12% Bis-Tris Gels (Invitrogen, Waltham, MA, USA). Wells were loaded with either 15 µL of sample-mix per well or 2 µL of the Precision Plus Protein Dual Color Standards (Bio-Rad laboratories, Inc., Hercules, CA, USA). The SDS PAGE ran at 200 V for 45 min. The proteins were transferred to an Immobilon-FL PVDF membrane (Merck Millipore, Ltd., Burlington, MA, USA) and blotted. The membranes were then incubated with the specific primary antibodies overnight. Next, membranes were transferred to a blocking buffer containing the secondary antibody, and incubated for 1 h at room temperature. The membranes were imaged with an Oddyssey CLx (Li-Cor, Inc., Lincoln, NE, USA) and the band strength was assessed using Image Studio Lite (Li-Cor, Inc.).

### 2.5. Protein-Network Analysis

The protein–protein interaction network analysis was performed using Cytoscape (v3.9.1) [21] with the STRING plugin (v1.7.0). The list of 25 overlapping DEGs was used as input and imported into Cytoscape. The minimum interaction score for the analysis was set to medium (0.4). Proteins with more than five interactions with other proteins were identified as hub proteins.

### 2.6. Gene Ontology Enrichment Analysis

The 25 DEGs were analyzed for enriched terms using STRING enrichment functions in the Cytoscape software and visualized using EnrichmentMAP (v3.3.3). The results were filtered to only include significantly enriched Gene Ontology (GO) biological processes (BPs). The significant threshold was set to *p* < 0.05.

## 3. Results

### 3.1. Transcriptomic Analysis of In Vitro and In Vivo Data

Using a combined criteria of FDR < 0.05 and absolute log_2_ fold change (abs(log_2_FC)) > 1, we identified a total of 1998 DEGs combining all the timepoints from the in vitro data, and 212 DEGs combining low and normal EF from the in vivo data. Out of these, 1496 and 126 were unique DEGs within the in vitro and in vivo data, respectively. As shown in the Venn diagram in Figure 1, 25 DEGs overlapped between the in vitro and in vivo data (Table 1). A detailed list of the 25 genes (FDR ≤ 0.05, abs(log_2_FC) > 1) is shown in Table A1 and Table A2.

### 3.2. Functional Assessment of the 25 DEGs

The corresponding proteins from the 25 DEGs were expressed in different cellular locations, including the extracellular space, the plasma membrane, and cytoplasm (Figure 2a,b). Relevant pathways associated with more than one of these genes are, for example, ‘Cardiomyocyte differentiation via bmp receptors’, ‘IGF-1 signaling’, ‘Cardiac hypertrophy signaling (enhanced)’, ‘Factor promoting cardiogenesis in vertebrates’, ‘Apelin cardiac fibroblast signaling pathway’, ‘ILK signaling’, and ‘Glucoticoid receptor signaling’. Five significant hypertrophy pathways were identified to be associated with the DEGs, including ‘Right ventricular hypertrophy’ (*p* = 4.31 × 10^−2^), ‘Concentric hypertrophic cardiomyopathy’ (*p* = 4.15 × 10^−7^), ‘Hypertrophy of the heart’ (*p* = 1.06 × 10^−4^), ‘Hypertrophy of the left ventricle’ (*p* = 4.95 × 10^−3^), and ‘Hypertrophy of cardiomyocytes’ (*p* = 6.07 × 10^−4^).

In this paper, we report on several novel DEGs, including *IGFB5*, *MFAP4*, *COL14A1*, *COL12A1*, *NES*, *THBS1*, *CCN1*, *KCNIP2*, *ACKR3*, *RASL1B*, *PLSCR4*, *FAM155B*, *PPP1RA*, *CLU*, *PDLIM7*, *DDAH1*, *IRS2*, *HSPA2*, and *ACTN1*. The association of these genes to any of the five hypertrophy pathways has not been shown before.

Using the IPA’s Tox function tool, we were able to significantly link 18 cardiac clinical pathology endpoints to the 25 overlapping DEGs (Figure 2c). Interestingly, the cardiac endpoint with the highest number of genes from our list of candidate hypertrophy biomarkers is the ‘Cardiac enlargement’ (*p* = 2.65 × 10^−5^, n = 7).

### 3.3. Protein Validation of the Selected DEGs

Seven of the DEGs were selected for protein validation, including *CCN1*, *CCN2*, *COL12A1*, *NES*, *NPPA*, *NPPB*, and *THBS1*. The selected candidates show upregulated gene expression levels in both in vivo and in vitro data. They were detected in biofluids, and are known to be secreted into the extracellular space.

The protein ‘Cellular Communication Network Factor 2’ (*CCN2/CTGF*) is known to be induced in heart failure and showed upregulation in the hypertrophic samples in both the in vitro and in vivo data. The protein analysis of the conditioned media from the in vitro samples showed a more than 3-fold increase in *CCN2/CTGF* (FC = 3.3, *p* = 0.0032) in the cultures stimulated with ET-1 for 24 h (150.5 ng/mL vs. 45.1 ng/mL) (Figure 3A). The gene expression data showed comparable results with a 3-fold upregulation at 24 h (Table A1).

Thrombospondin 1, which is a glycoprotein that is involved in cell–cell and cell–matrix interactions, was first analyzed with the ELISA method, but due to difficulties in detecting a signal, a quantitative Western blot analysis was performed instead. The results showed a significant 2.3-fold increase at 24 h (*p* = 0.02) (Figure 3B and Appendix A). From the gene expression data, a significant upregulation was also observed at 24 h (FC = 2.2) (Table A1).

The natriuretic proteins ANP and proBNP, known markers for cardiac hypertrophy, were both upregulated significantly in the ET-1 stimulated cultures. The ANP level in ET-1 induced sample was more than 2-fold (*p* = 0.006) higher compared to the control at 24 h (Figure 3C). A much higher concentration of proBNP (>9-fold (*p* = 0.002)) was detected in ET-1-stimulated samples compared to the controls at 24 h (Figure 3D).

No proteins of CCN1, COL12A1, and NES were detected using ELISA or Western blot (data not shown).

### 3.4. Protein Interaction Network Analysis

A STRING protein–protein interaction (PPI) network analysis was performed to investigate any known protein interactions among the 25 DEGs identified from intersecting the in vitro and in vivo data. The PPI enrichment shows that the derived network has significantly more interactions than what is expected by random analysis (*p*-value 6.34 × 10^−7^). Interestingly, *CCN2/CTGF* protein was detected as the central hub node in the network with seven interactions to other differentially expressed gene products. The color of the nodes indicates the magnitude of the log_2_FC differences in gene expression. High similarities in FC magnitude between the control and hypertrophic samples in both the in vitro and in vivo data were observed. This suggests that the hypertrophic response in the in vitro model mimics to a high degree the hypertrophic response of CMs in vivo. However, some genes, e.g., *IGFBP5*, *COL14A1*, *MFAP4*, *ACKR3*, *PLSCR4*, and *CLU*, show a conflicting differential expression direction in the in vitro and in vivo data (Figure 4).

### 3.5. Gene Ontology (GO) Enrichment Analysis

The GO enrichment analysis was performed to explore enriched biological process (BP) terms identified for the 25 DEGs. In total, 28 GO-BP terms were significantly enriched. The five most enriched terms were ‘Regulation of systemic arterial blood pressure’, ‘Positive regulation of cardiac muscle contraction’, ‘Anatomical structure morphogenesis’, ‘Regulation of system process’, and ‘Positive regulation of reactive oxygen species metabolic process’. All the enriched terms are visualized in Figure 5.

## 4. Discussion

In this study, we identified candidate cardiac hypertrophy biomarkers that show significant differential expression both in an hiPSC-CM hypertrophy in vitro model and in human hypertrophic heart biopsies. We utilized published transcriptomic data from Aggarwal et al. (2014) and Johansson et al. (2020). Johansson et al. performed transcriptional characterization of an in vitro hypertrophy model in their study, and Aggarwal et al. compared mRNA expressions using PCA on overlapping DEGs to highlight differences between in vitro and in vivo data. In the present study, we further analyzed these datasets, but now focused on identifying similarities instead of differences between the in vitro model and in vivo data. The overlapping DEGs were further validated on a protein level. These results provide novel data, showing how the stem-cell-based hypertrophy model mimics the hypertrophy of the heart.

Supporting our findings, previously known biomarkers of hypertrophy, such as *NPPA* and *NPPB*, also showed a significant differential expression in our datasets. In this paper, we identified and validated two potential cardiac hypertrophy markers, *CCN2*/*CTGF* and *THBS1*.

The PPI analysis identified *CCN2/CTGF*, which encodes for the protein ‘connective tissue growth factor’, as a central hub node, indicating an important role in the progression of cardiac hypertrophy. The gene expression of *CCN2*/*CTGF* was also found to be highly upregulated in both the in vitro and in vivo data. Moreover, the corresponding protein was also significantly upregulated in the conditioned media from the ET-1-treated hiPSC-CMs. Increased levels of *CCN2/CTGF* have also been observed in conditions such as myocardial infarction, hypertension, and diabetes [22,23,24]. Combining that information with its known profibrotic properties makes it a plausible contributor to the development and progression of cardiac hypertrophy. ET-1 stimulation of rat CMs has previously been shown to upregulate *CCN2/CTGF*. Additionally, it has been shown that direct treatment by *CCN2/CTGF* protein stimulates the hypertrophic growth of the cells in rat CMs. However, the expression of known hypertrophic markers, such as BNP and ACTA1, were not elevated under these conditions due to the modulation of different signaling pathways compared to the ET-1 stimulation of CMs, suggesting that *CCN2/CTGF* protein alone is not sufficient to induce the pathological form of cardiac hypertrophy [25].

There are several studies indicating the role of *CCN2/CTGF* protein as a biomarker for cardiovascular diseases and report that plasma levels of *CCN2/CTGF* protein correlate with the plasma levels of the known hypertrophy marker BNP in patients with heart failure. It has also been suggested that *CCN2/CTGF* protein could be a biomarker for comparing the severeness of heart failure [26,27,28]. Moreover, a recent study comparing patients that underwent septal myectomy due to cardiac hypertrophy to structurally normal hearts that were harvested following non-cardiac-related deaths showed that *CCN2/CTGF* was significantly upregulated and appeared to be a key mediator of myocardial fibrosis [29]. This study supports the potential role of *CCN2/CTGF* as a biomarker of cardiac hypertrophy.

The *THBS1* gene encodes for a binding glycoprotein thrombospondin 1 (TSP-1). It is associated with angiogenesis, cell migration, and cardiac remodeling, which are hallmarks of cardiac hypertrophy. The TSP-1 protein has been shown to be both pro- and anti-angiogenic [30,31,32]. It has also been shown that *THBS1* promotes follicular angiogenesis and hypertrophic scar fibroblasts [33,34]. The overexpression of TSP-1 induces lethal cardiac atrophy (a reduction in heart muscle mass) in mice, and the TSP-1 knocked-out mice develop a severe cardiac hypertrophy during the stimulation [35]. In contrast, we found that *THBS1* is overexpressed in both the cell-based hypertrophy model and in the cardiac biopsies from patients with cardiac hypertrophy. These results highlight the differences in the cardiovascular system in animals and in humans and suggest that the development of a relevant in vitro model that represents the mechanism of cardiac hypertrophy in humans is of great importance. The TSP family has several functional domains that interact with different proteins and receptors. TSP-1 is pro-angiogenic when interacting with α9β1 integrin [30], but notably, the opposite effect occurs when it interacts with CD36 and CD47 [31,32]. The expression of TSPs increases under cardiac stress [36] and results from a study on canine and murine models showed that the elevation of TSP-1 during myocardial injury helps to protect the heart and prevent the expansion of infarctions [37]. Another study suggests that the increase in TSP-2 (an important paralog of TSP-1) is a sign of cardiac hypertrophy that is prone to develop into heart failure [38].

In an animal disease model of hypertrophic cardiomyopathy with a significant enlargement of the left ventricle and severe heart failure, it was shown that both *THBS1* and *CCN2/CTGF* were significantly upregulated in the diseased condition [39]. The analysis of *THBS1* and *CCN2/CTGF* in both CMs and non-CM isolated from the mice showed that there was an upregulation of *CCN2/CTGF* in both CMs and non-CMs. However, for *THBS1*, the upregulation was only observed in the CMs but not in the non-CMs population. This result is in line with another study, for which the results suggest that CMs accounts for the upregulation of the *THBS1*-CD47 axis in left ventricular heart failure [40]. *CCN2/CTGF* was also analyzed on a protein level, and the results showed a plasma level more than double when compared to the control mice. Interestingly, it was shown that treating neonatal ventricular cardiomyocytes (NVCM) with recombinant *CCN2/CTGF* protein resulted in a rapid upregulation of *THBS1* [39].

We identified *THBS1* as a biomarker candidate for cardiac hypertrophy, since it is overexpressed in both the hiPSC-based hypertrophy model and in biopsies from patients with cardiac hypertrophy. Little is known about the potential role of *THBS1* in cardiac hypertrophy and more research on the interactions between *THBS1* and the different receptors and the mechanism underpinning the TSP protein expression is required to provide a better understanding of the role of TSP-1 in cardiac hypertrophy and cardiovascular diseases.

The *NPPA* and *NPPB* genes are established cardiac hypertrophy biomarkers. These paralog genes encode natriuretic peptide cardiac hormones, the ANP and BNP proteins. They mediate cardiovascular homeostasis through the regulation of natriuresis, diuresis, vasorelaxation, and renin inhibition [41,42,43,44]. ANP and BNP are highly expressed during the development of the ventricles, but strongly downregulated after birth [45,46,47]. Studies show a reactivation of the two hormones in ventricular CMs during heart failure and cardiac hypertrophy [45,48,49,50]. These two genes were upregulated in both the in vitro ET-1-induced cardiac hypertrophy model and in the hypertrophic heart biopsies. In our in vitro model, *NPPA* was upregulated at all time points during ET-1 stimulation, while the *NPPB* was upregulated at 8 h, 24 h, and 72 h (but not at 48 h and 96 h). In the in vivo heart biopsy data, the *NPPA* and *NPPB* genes were upregulated only in the hypertrophic hearts with a low EF. This suggests that *NPPA* and *NPPB* become more relevant at the later stage of cardiac hypertrophy. Further investigations on the characteristics of cardiac hypertrophy and the progression of this severe condition are still needed to further refine the disease models.

The network analysis of the overlapping 25 DEGs that were significantly upregulated in hypertrophic samples showed interesting interaction patterns, and of specific note is that *CCN2/CTGF* was identified as a hub protein in the network with interactions for seven of the other 25 DEGs. This supports the fact that *CCN2/CTGF* is an important protein in cardiac hypertrophy development and likely of high relevance for benchmarking hypertrophy progression.

The GO analysis also revealed that most of the enriched terms are of high relevance to cardiac hypertrophy development and indicates that data from the in vitro model reflect the in vivo model in important aspects. It was also confirmed in the cardiac clinical endpoint analysis, which found cardiac enlargement as the most significant endpoint. The expression profiles of several hypertrophy genes, such as NPPA and NPPB, are highly similar for both in vitro and in vivo data. The in vivo data used in this study were generated using the Affymetrix microarray platform with predefined probes, which, to some extent, limited the possibilities to identify novel transcripts. For future studies, it would be of interest to analyze data from patient biopsies using the RNA-seq technology, which can identify novel transcripts and allow for a better comparison with the in vitro model. Additionally, blood samples from patients with cardiac hypertrophy would be important to include in future studies to verify if the suggested candidate biomarkers are detectable in blood samples of patients with cardiac hypertrophy. Despite the current limitations, the results from this study demonstrate that in vitro models of cardiac hypertrophy have a great potential to be effective tools for novel drug-screening approaches and for the development of novel therapies for the treatment of cardiovascular diseases.

## 5. Conclusions

The differential expression comparison between the hiPSC-CM-based in vitro model and hypertrophic cardiac biopsies allows potential cardiac hypertrophy biomarkers to be identified and validated. The functional enrichment analysis and the PPI analysis revealed that the overlapping 25 DEGs were associated with cardiac functions and cardiac hypertrophy. Most of the overlapping genes were differentially expressed in the same direction in the in vivo and in vitro samples, except for six genes (*ACKR3*, *CLU*, *COL14A1*, *IGFBP5*, *MFAP4*, and *PLSCR4),* which were downregulated in the in vitro model but upregulated in the hypertrophic cardiac biopsies. In the study, *CCN2/CTGF* and *THBS1* were identified as interesting candidate cardiac hypertrophy biomarkers that are expressed in both the ET-1-induced cardiac hypertrophy in vitro model and the hypertrophic cardiac biopsies at both gene and protein levels. Additional research is needed to be able to reveal the specific roles of the identified biomarkers in cardiac hypertrophy and to validate their potential applications in a clinical setting.

## Figures and Tables

**Figure 1 life-12-00726-f001:**
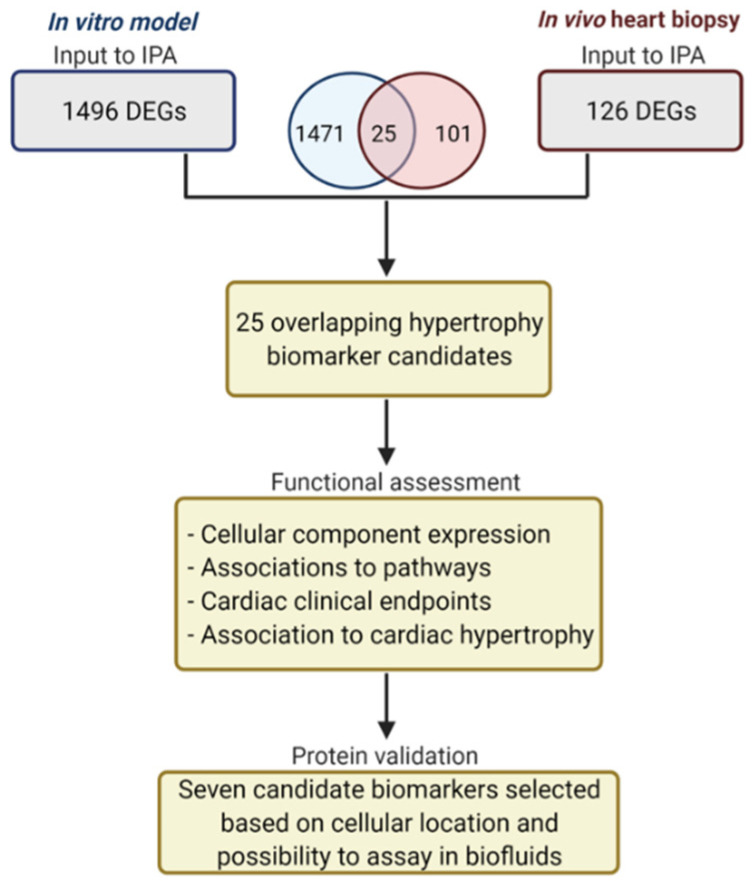
Overview of the candidate hypertrophy biomarker identification and validation process.

**Figure 2 life-12-00726-f002:**
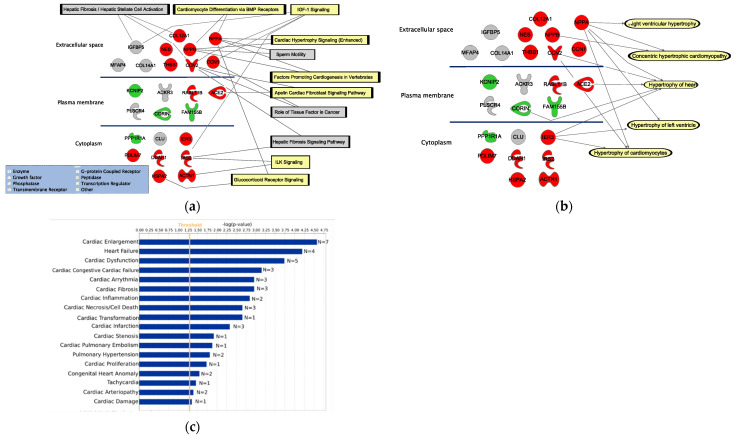
Functional assessment of the 25 DEGs. (**a**) Pathways associated with two or more of the DEGs (pathways associated with the hypertrophy condition are in yellow). Red represents DEGs that are upregulated in both in vitro and in vivo data and green represents downregulated. Grey indicates that the DEGs are regulated in the opposite direction in the in vivo and in vitro data. (**b**) Association of the DEGs to cardiac hypertrophy. (**c**) Association of the DEGs to cardiac clinical endpoints.

**Figure 3 life-12-00726-f003:**
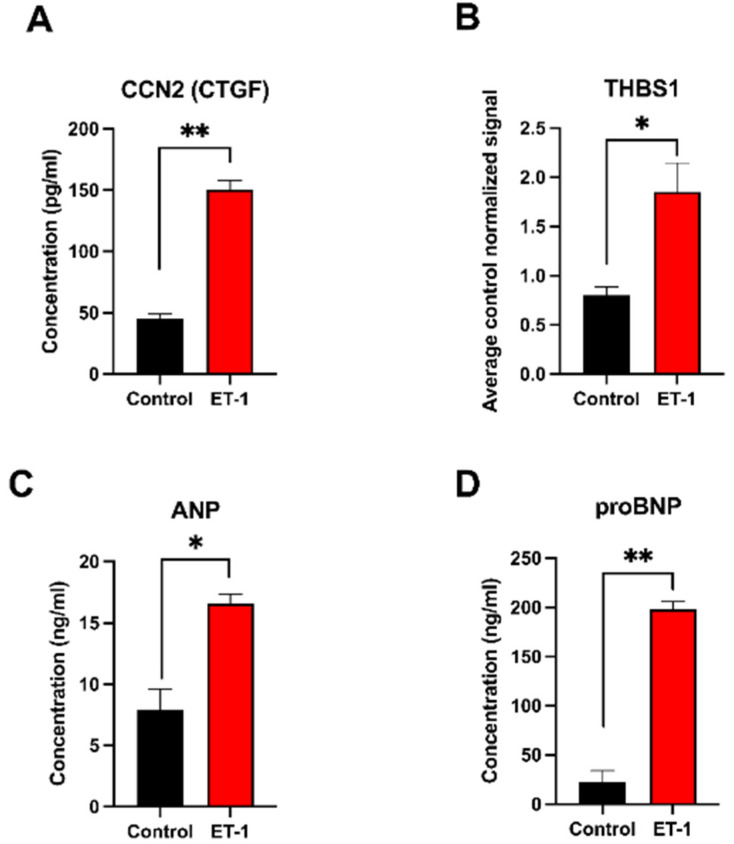
Differentially expressed proteins in the conditioned media analyzed with ELISA and Western blot after 24 h of ET-1 stimulation. (**A**) Concentration of the *CCN2* (*CTGF*) protein. Y-axis shows the concentration in pg/mL. (**B**) Expression levels of the *THBS1* protein at 24 h measured with quantitative Western blot. The Y-axis shows the average control normalized signal (representing expression level). (**C**,**D**) Concentration of the ANP and proBNP proteins, respectively. Y-axis shows the concentration in ng/mL. Standard deviation (SD) is shown as error bars (n = 3); * = *p* < 0.05; ** = *p* < 0.01.

**Figure 4 life-12-00726-f004:**
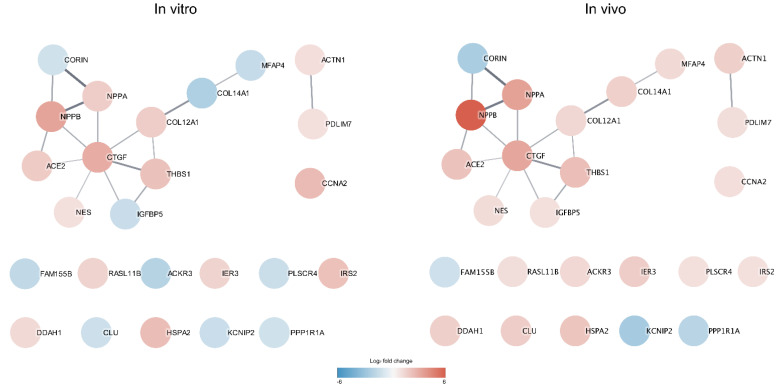
STRING protein–protein interaction network (PPI) was generated by importing the 25 overlapping DEGs between the in vitro and in vivo data into Cytoscape and the STRING plugin. The lines between the proteins indicate an interaction. The thicker the line is, the more confident is the evidence for this interaction between the proteins. The color of the proteins indicates the FC levels observed in the in vitro (**left** PPI network) and in vivo data (**right** PPI network). Red colors represent upregulation and blue colors represent downregulation.

**Figure 5 life-12-00726-f005:**
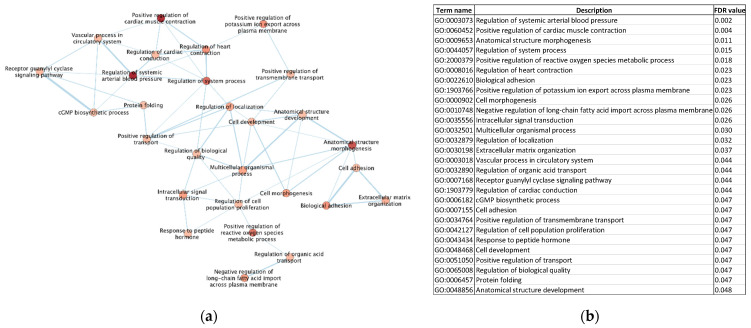
Gene Ontology (GO) enrichment map. (**a**) The 25 overlapping DEGs were analyzed for enriched GO terms (Biological Process (BP)). The color of the circles, representing a GO-BP term, corresponds to the *p*-value (FDR). The thickness of the lines between terms indicates how closely related the terms are to each other. The thicker the line, the more closely related. (**b**) A table with the FDR-values of the terms in the GO enrichment map in (**a**).

**Table 1 life-12-00726-t001:** The 25 overlapping DEGs were significantly differentially expressed (FDR ≤ 0.05, abs(log_2_FC) > 1) in both the in vitro and in vivo hypertrophy datasets. This table shows their detectability in biofluids and their expression levels in the in vitro models and in the hypertrophic cardiac biopsies. Each square represents an upregulated (

), downregulated (

), or non-differentially expressed gene (

) at the different time points (8, 24, 48, 72, and 96 h; n = 3 each) in the in vitro samples and in the cardiac samples with a normal EF (n = 3) or low EF (n = 3).

Gene Symbol	Blood	Plasma/Serum	Urine	Not Detected in Biofluids	In Vitro Expression(8–96 h)	In Vivo Expression(Normal and Low EF)
*ACE2*			x		    	 
*ACKR3*	x	x			    	 
*ACTN1*	x	x	x		    	 
*CCN1*	x		x		    	 
*CCN2*	x		x		    	 
*CLU*	x	x	x		    	 
*COL12A1*			x		    	 
*COL14A1*			x		    	 
*CORIN*				x	    	 
*DDAH1*			x		    	 
*FAM155B*	x	x			    	 
*HSPA2*				x	    	 
*IER3*				x	    	 
*IGFBP5*	x	x	x		    	 
*IRS2*	x	x			    	 
*KCNIP2*				x	    	 
*MFAP4*			x		    	 
*NES*	x	x			    	 
*NPPA*	x	x			    	 
*NPPB*	x	x			    	 
*PDLIM7*				x	    	 
*PLSCR4*				x	    	 
*PPP1R1A*				x	    	 
*RASL11B*				x	    	 
*THBS1*	x	x	x		    	 

## Data Availability

This study is based on two trancriptomics datasets, which are available for download at ArrayExpress (https://www.ebi.ac.uk/arrayexpress/, accessed on 4 April 2022) accession numbers: E-MTAB-11030 and E-MEXP-2296.

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
