# Peer review of "Data Mining Identifies CCN2 and THBS1 as Biomarker Candidates for Cardiac Hypertrophy"

_life, 2022, doi:10.3390/life12050726_

Round 1

Reviewer 1 Report

The authors present an interesting follow-up study to their previously published paper Johansson et al., 2020. This is a good example of how data from a previous study can still be utilized for new insights. I think the comparison of in vitro with in vivo generated data is very important and has been adequately discussed by the authors, especially concerning different findings in different species and how this might lead to opposing findings. The authors emphasize strongly that upregulation of CCN2 and THBS1 are novel markers of cardiac hypertrophy, which is somewhat conflicting with several previous reports in animal models and human studies (Tsoutsman et al. 2013; Patel et al. 2021). Therefore, I think the authors should further specify in their manuscript how and why they define their findings as novel? It seems that the novelty is the direct comparison of in vitro data with in vivo data in the same study, whereby indirectly this has and is being done by most disease modeling studies that use hiPSC-derived cardiomyocytes nowadays. Therefore, before final publication I have the following suggestions for the authors:

  • Soften the message of novelty concerning biomarker findings and work out “real novelties”, focusing on the experimental approach and the THBS1 upregulation, which seems to have been associated with fibrotic scarring in HCM patients and animal models. 
  • Interesting for upregulation of THBS1 might be the fact that it seems to be mainly expressed by fibroblasts in the heart. Therefore, it is important that the authors comment on a potential “contamination” of their in vivo and in vitro analysis with other non-cardiomyocyte cell types and if that has been considered for interpretation of their results.
  • In this line soften the title of the manuscript erasing the word “novel”.

Minor Comments:

  • Please supply an image of the western blot membrane corresponding to figure 3B.
  • Please indicate throughout the manuscript technical and biological replicates were adequate.

Patel, V., P. Syrris, C. Coats, J. Lucena, E. Lara-Pezzi, P. Garcia-Pavia, and P. M. Elliott. 2021. “Genetic Regulation of Myocardial Fibrosis in Hypertrophic Cardiomyopathy.” European Heart Journal 42 (Supplement_1). https://doi.org/10.1093/eurheartj/ehab724.1778.

Tsoutsman, Tatiana, Xiaoyu Wang, Kendra Garchow, Bruce Riser, Stephen Twigg, and Christopher Semsarian. 2013. “CCN2 Plays a Key Role in Extracellular Matrix Gene Expression in Severe Hypertrophic Cardiomyopathy and Heart Failure.” Journal of Molecular and Cellular Cardiology 62 (September): 164–78.

Reviewer 2 Report

The study of Johansson et al used data analysis of pre-existing datasets to identify CCN2 and THBS1 as potential biomarkers for cardiac hypertrophy. 

This reviewer was left confused. The authors used two RNA-seq datasets. One they have published previously (Biol Open (2020) 9 (9): bio052381) and another published by another group (PLoS One (2014) Sep 25;9(9):e108051). Both of the previous studies analyzed the RNA-seq thoroughly. As such, the need for a new analysis is unclear. What would help in this regard is a discussion of the findings of both papers, what this new analysis hopes to achieve and how this new analysis differs from the previous ones. All of this missing and needs to be added. It is especially pertinent because THBS1 has already been mentioned in the PloS One study as a novel finding. 

Table 1 does not help the authors' cause and needs some work. No N per sample/time-point is mentioned. Looking at the table many genes are differentially regulated at a single-time point. This is unlikely to be biologically relevant and suggests false positives. Did the authors do any tests to determine the likelihood of false positives? 

Is Gene Ontology appropriate on a set of 25 genes? This seems too small for the assumptions of Gene Ontology to remain valid.

The rationale for cutting down 25 candidates to 7 is missing.

Round 2

Reviewer 2 Report

This reviewer has no further comments.